# Capillary Flow-Driven and Magnetically Actuated Multi-Use Wax Valves for Controlled Sealing and Releasing of Fluids on Centrifugal Microfluidic Platforms

**DOI:** 10.3390/mi13020303

**Published:** 2022-02-16

**Authors:** Snehan Peshin, Derosh George, Roya Shiri, Lawrence Kulinsky, Marc Madou

**Affiliations:** 1Department of Mechanical and Aerospace Engineering, University of California, Irvine, CA 92697, USA; lkulinsk@uci.edu (L.K.); mmadou@uci.edu (M.M.); 2Department of Civil and Environmental Engineering, Princeton University, Princeton, NJ 08540, USA; deroshmekkattu@gmail.com; 3Autonomous Medical Devices Incorporated, Santa Monica, CA 90404, USA; shirir@uci.edu

**Keywords:** centrifugal microfluidics, microfluidic valving, point-of-care diagnostics, Lab-on-CD

## Abstract

Compact disc (CD)-based centrifugal microfluidics is an increasingly popular choice for academic and commercial applications as it enables a portable platform for biological and chemical assays. By rationally designing microfluidic conduits and programming the disc’s rotational speeds and accelerations, one can reliably control propulsion, metering, and valving operations. Valves that either stop fluid flow or allow it to proceed are critical components of a CD platform. Among the valves on a CD, wax valves that liquify at elevated temperatures to open channels and that solidify at room temperature to close them have been previously implemented on CD platforms. However, typical wax valves on the CD fluidic platforms can be actuated only once (to open or to close) and require complex fabrication steps. Here, we present two new multiple-use wax valve designs, driven by capillary or magnetic forces. One wax valve design utilizes a combination of capillary-driven flow of molten wax and centrifugal force to toggle between open and closed configurations. The phase change of the wax is enabled by heat application (e.g., a 500-mW laser). The second wax valve design employs a magnet to move a molten ferroparticle-laden wax in and out of a channel to enable reversible operation. A multi-phase numerical simulation study of the capillary-driven wax valve was carried out and compared with experimental results. The capillary wax valve parameters including response time, angle made by the sidewall of the wax reservoir with the direction of a valve channel, wax solidification time, minimum spin rate of the CD for opening a valve, and the time for melting a wax plug are measured and analyzed theoretically. Additionally, the motion of the molten wax in a valve channel is compared to its theoretical capillary advance with respect to time and are found to be within 18.75% of the error margin.

## 1. Introduction

Centrifugal microfluidics platforms are becoming widely implemented for various applications due to the possibility of integrating sample preparation steps (such as blood plasma separation [1]) with assays. Additionally, centrifugal fluidics features a reliable, controllable, and compact pumping mechanism that facilitates rapid mixing of reagents [2], short response time, and improved assay sensitivity. Because fluid propulsion is enacted by the centrifugal forces on a rotating disc, there is no input/output tubing, thus avoiding contamination of the sample by the surroundings. Microvalves are critical elements in any CD fluidic design to selectively open or close microfluidic channels to properly route the fluids as well as to enable serial dilution and to release or isolate the contents of specific reservoirs.

Microvalves can be categorized into “active” and “passive” valves. Passive valves do not require external peripheral devices to operate. For instance, passive capillary valves are opened simply by increasing the angular velocity of the microfluidic CD as the centrifugal force overcomes the surface tension force [3]. Passive valves include capacitance switch types [4], capillary burst [5], dissolvable film-based [6], hydrophobic [7,8,9], siphon [10,11,12,13], and phase change types [14]. In contrast, active valves require external peripheral devices for actuation. Examples of peripheral means to actuate active valves include lasers to burn through a film [15] and magnets that act on metal plugs to open or close [16]. Other methods include manual hand pressure, electrical fields, and heating [2,14,17,18]. Typically, active valves are more reliable than passive valves because the latter depend on the constancy of material properties, the absence of local defects, and negligible manufacturing variations. Increased reliability of the active valves often comes at the expense of additional required peripheral pieces of equipment, therefore increasing cost and complexity and decreasing portability of the systems.

Active wax valves belong to a class of phase change-based valves that are actuated by heat [14] and can be combined with other external actuators, like a magnetic field [14,19]. Adding ferrofluid to the wax reduces the melting point of the wax [14,19] which helps in the phase-change process. The ferroparticles enhance the absorption of the incident light, facilitating faster actuation of the valve. However, most of the existing designs of such valves [14] only enable one-time action and do not allow repeated opening and closing. Following is the advantages of our design over previously reported designs [14]:The previous wax valves [14] can only be used once, either from the off to on configuration or from on to off configuration.There are twice the number of valves required for performing the same function as done by our design due to the need for separate valves for on and off functions.Other designs require more space on the CD, which makes the fabrication more expensive and laborious.

Additionally, a typical implementation of wax valves adds to the complexity and expense of microfluidic platforms, because for reliable operation of wax valves, micro-actuator heating elements [7] and even vacuum sources [3] are required. Existing wax valves would benefit from further improvements to allow for their more reliable and reversible operation.

The present work describes two new types of phase change-based ferrowax valves—a multiuse capillary flow-driven wax valve [20] and a reversible magnetic wax valve. These two types of valve designs consist of a wax chamber containing ferrowax connected to a valving channel. To close the valving channel with wax, the wax chamber and the valve channel are heated. Under the action of the capillary force, the molten wax flows into the valving channel and seals it. Opening of the sealed channel is achieved by locally melting the wax inside the channel (but not inside the wax reservoir) and spinning the disc to flow the molten wax plug downstream, clearing the channel. As long as there is wax present in the wax chamber, it is possible to melt it and re-seal the fluidic channel, enabling multiple closings and openings of the fluidic conduit (Figure 1a,b). In the magnet-actuated wax valve, a magnet reversibly pulls the molten wax in and out of the valving channel to switch it on or off (Figure 1c,d). The magnetic ferrofluid that is mixed with the wax helps reduce the melting point of the wax as well as help actuate it with a magnet. The term multi-use as applied to wax valves discussed in this work indicates the use of the valve to open and close the fluidic channel multiple times for the same chemical or biological assay rather than using the same valve for different assays run on the same disc. Typically microfluidic platforms are not reused for different assays, but are disposed. The wicking of the ferrowax between the wax reservoir and the fluidic channel was numerically simulated and analyzed to optimize the valve design. These optimized multi-use valves were fabricated and tested as described in the following sections. Although ferrofluid is a critical component for magnet-actuated wax valve, it is not necessary to use ferrowax for capillary wax valve described above, and using a regular wax rather than ferrowax is sufficient. One side benefit of using the ferrowax instead of the regular wax for a non-magnetic valve is faster absorption of the heat and thus faster melting time for solid wax.

## 2. Materials and Methods

### 2.1. Characterization of the Ferrowax

The ferrowax used in our experiments is prepared by mixing paraffin wax (327204, Sigma-Aldrich, Inc., St. Louis, MO, USA) with Ferrofluid (EFH1, 60 cc, Ferrotec, Santa Clara, CA, USA) in the ratio of 2:1 by mass and stirring the mixture at 65 °C (hotplate) for 12 h.

Both surface tension coefficient of the ferrowax and its contact angle with the acrylic surface and with the adhesive surface were measured. The surface tension of the ferrowax is calculated by the pendant drop methodology [21] wherein a Hamilton syringe filled with the ferrowax is heated (Heater Kit-1LG, New Era Pump Systems, Inc., Farmingdale, NY, USA) until the ferrowax melts and a drop is suspended from the syringe’s needle. The image of the wax droplet is captured using a high-speed camera (18W15, Kron Technologies, Burnaby, BC, Canada) (Figure 2a and Appendix A). The captured image is analyzed with the ImageJ software to deduce the surface tension coefficient. The calculated surface tension coefficient for the ferrowax is 0.051 +/− 0.005 N/m.

The contact angle is measured using the setup shown in the Figure 2b. A camera (MA1000, Amscope, Irvine, CA, USA) captures a side view image of the molten ferrowax droplet on the polyacrylic and adhesive surfaces. These images are analyzed with the ImageJ program (National Institute of Health, Bethesda, MA, USA) using the software’s in-built contact angle plugin [22]. The calculated contact angle value for the ferrowax is 15 +/− 0.5° with the adhesive surface (single-sided pressure-sensitive adhesive, 9795R, 3M, Saint Paul, MN, USA), and it is 19 +/− 0.5° degrees with the acrylic surface (8589K43, Clear Scratch- and UV-Resistant Acrylic Sheet, McMaster-Carr, Elmhurst, IL, USA). The literature-reported values for density and viscosity of the ferrowax are 760 kg/m^3^ and 0.00031 kg/m·s, respectively [23]. These values are documented in Table 1.

### 2.2. Design and Fabrication of Ferrowax Valves

#### 2.2.1. Valve Fabrication

The valves on the disc are designed in the 3-D Solidworks modeling software from Dassault Systèmes, Vélizy-Villacoublay, France and machined using a CNC machine (Tormach 440 PCNC CNC milling machine, Tormach, Waunakee, WI, USA) in a polyacrylic (PMMA) sheet (8589K43 Clear Scratch- and UV-Resistant Acrylic Sheet, McMaster-Carr, Elmhurst, IL, USA). A single-sided pressure-sensitive adhesive tape (9795R, 3M, Saint Paul, MN, USA) is used to seal the channels and chambers. These layers are depicted in Figure 2a. The ferrowax mixture is brought down to room temperature and placed inside the designated wax chamber.

Alternatively, the wax can be injected in molten form using a heated Hamilton syringe (Figure 3a). The valve assembly sequence is presented in Figure 2b,c. The top layer of this assembly consists of the single-sided adhesive sheet that contains the openings for vent holes, alignment holes, and openings for loading the wax in the wax chamber. The bottom layer is an acrylic layer machined to provide for the upstream chamber, the wax inlet chamber, the valve channel, and the downstream chamber. The layers are pressed together using a roller (B07YDNKSH6, Akiro, Shenzhen, Guangdong, China, Amazon, USA) to activate the pressure-sensitive adhesive that binds the layers together.

#### 2.2.2. Capillary Force-Driven Wax Valve Operation

There are two ways of fabricating a capillary-driven wax valve based on the method in which wax is deposited in the wax reservoir. In one method, a physical piece of solid ferrowax (0.2 mg) is placed into the wax reservoir through an opening in the adhesive layer. This approach is schematically illustrated in Figure 3d. Either the CD is heated (using a hotplate) to 90 °C for up to 1 min or the wax is heated with a laser (500 mW, Q-Baihe, 808D-500–9.0 mm-IR, Amazon, USA, for up to 5 s) until it melts. The molten wax under the influence of the capillary action flows into the fluidic channel. When the wax reaches the channel, the CD is cooled down (by placing it in a refrigerator for 5 s at 4 °C) where the ferrowax solidifies and blocks the fluidic channel.

Instead of depositing a solid piece of wax, it is also possible to place molten wax into the wax reservoir by using a heated Hamilton syringe. This is illustrated in Figure 2a. The molten wax is displaced from the syringe into the wax reservoir through the opening in the film above the wax reservoir. Similar to the technique described above, the molten wax, under the influence of the capillary force, flows into the fluidic channel and the fluidic channel closes when the wax cools down and solidifies. This molten wax deposition technique and flow of the wax towards the fluid channel with subsequent blocking of the channel is presented in Figure 4 and Appendix A.

As shown in Figure 4 and Appendix A, the molten wax overflows into the circular fluidic chambers. In order to prevent this, we can have a wedge angle greater than 180 degrees such that the wax cannot move into the chamber by the formation of capillary burst valve.

#### 2.2.3. Magnet Wax Valve Fabrication

The magnet-actuated wax valve was compared with the capillary force-based wax valve in Figure 1c,d. The magnetic force (rather than the capillary force) is the driving force for the motion of the ferrowax. To fabricate this valve, a solid piece of ferrowax (0.2 mg) is again placed inside the wax chamber through the opening in the film. The CD is then heated above the melting point of the wax using a hotplate (90 °C for up to 1 min), or a laser (500 mW for up to 5 s). After the wax is molten, a magnet (Grade 5 ferrite magnet, B075RZ82HP, LLCC, Amazon, Seattle 98109, WA, USA), 2 mm thick with a diameter of 6 mm [24], is placed over the wax chamber and advanced towards the fluidic channel and the ferrowax follows the magnet to flow into the channel. Once the channel is filled, the CD is cooled down (by placing it at 4 °C in the refrigerator for about 5 s) to solidify the wax.

### 2.3. Simulation of Capillary Wax Valves

Finite element analysis (FEA) of the actuation of capillary wax valves was performed using ANSYS Fluent software (ANSYS Inc., Canonsburg, PA, USA). Transient two-phase flow simulation of the molten ferrowax and air was implemented (as shown in Figure 5a). In Figure 5a, the initial condition for this simulation is depicted where the volume fractions of the ferrowax and the air are kept at 1 and 0, respectively, in the wax chamber (the grey region) and, correspondingly, 0 and 1 outside the wax chamber (the green region). In Figure 5b, the inlet (atmospheric pressure vent) is located at the top surface of the wax chamber (in red). Similarly, in Figure 5c, the two outlets (atmospheric pressure vent, in red) are located at the bottom surfaces of the two chambers (inward chamber and outward chamber). The adhesive boundary is defined in Figure 5d, and the experimental value for the wax contact angle of 15 degrees is assigned for this surface. For the acrylic boundary (Figure 5e), a measured wax contact angle of 19 degrees is used. The position of the capillary wax valve on the disc is presented in Figure 5f, where the “inward chamber” is located closer to the center of the disc and the “outward chamber” is located further away from the center of the disc. The fluidic channel has a radial orientation, and it connects the “inward” and the “outward” chambers. The centrifugal force pushes the fluid from the inward chamber towards the outward chamber. A contact mesh matching is performed between the intersection of the wax region and the air region to establish the common nodes at the interface and to ensure that flow is continuous.

The small contact angles between the wax and the contacting material (either 15° or 19°) result in a concave liquid meniscus that advances from the wax reservoir towards the junction with the valve channel. The propulsion of the wax meniscus by capillary force can be seen in the simulation results in Figure 6a–d (and in Appendix A). These simulation results match well with the observed experimental outcomes presented in Figure 4 and discussed in previous sections.

Present simulation results refer to wax valve without the adjoining fluid reagents. The actual melting and propagation of the wax takes longer due to the presence of heat-absorbing fluid. Qualitatively the valve action proceeds as expected as evidenced by experimental results.

To open a normally closed valve channel, the ferrowax in the channel is heated again and the disc is spun to push the wax into the outward chamber by the centrifugal force. This action is simulated as a two-phase laminar flow in ANSYS Fluent. The results of these 3D simulations are presented in Figure 6e,f, where the action of the centrifugal force at an angular velocity of 6500 rpm is simulated.

### 2.4. Experimental Setup

In Figure 2c, we present a schematic of the experimental setup used during the evaluation of the wax valves. This setup consists of a spin chuck (machined in the lab), a motor (SM3450D, Motion USA, Columbus, OH, USA), a BLDC Servo Motor Controller (EZSV23/EZSV17, AllMotion, Union City, CA, USA), a strobe light (DT-311A, Shimpo, Lynbrook, NY, USA), a camera (acA800-510uc, Basler AG, Ahrensburg, Germany), and an optical sensing unit (D10DPFP, Banner Engineering Corp., Minneapolis, MN, USA). The motor-driven spin chuck holds the CD, controlled by the controller that receives input for the sequences of the angular velocities and accelerations from the user interface of the computer. A stationary optical sensing unit helps keep track of the actual rotational frequency of the disc by shining a light on the rotating CD and detecting the light reflected by a reflecting tape on the top surface of the CD when it passes beneath the optical sensor. Every time the light is detected, the strobe light and camera are turned on and one picture per revolution of the disc is taken. This frame-by-frame sequence is recorded as a video using a screen recording software Bandicam (Bandicam Company, Irvine, CA, USA) to analyze the fluidics on the CD.

## 3. Results and Discussion

For each of the experiments discussed below, the radial position of the valve is 20 mm from the center of the disc, the fluidic channel is 0.5 mm deep and 0.5 mm wide, and the wax chamber depth is 1 mm. The shape of the extruded cut of the wax reservoir is shaped like a trapezoid prism with shorter parallel surface of 1.2 mm long, distance between the parallel sides of 3 mm, width w for our reservoir is 2 mm, height h is 0.25 mm, and the valve convergence angle is 40° (where not mentioned).

### 3.1. Operation of the Capillary Wax Valve

#### 3.1.1. Functional Tests of Capillary Wax Valves

The solubility of ferrowax was tested by immersing a 1 mg piece of solid ferrowax in 10 mL of water for a week. The water was removed, and the weight of the solid wax was checked after the test. No change in weight was observed, indicating the compatibility of using wax with aqueous working fluids and reagents.

The main function of the capillary-driven wax valve is to close the valve without any leakage, open the valve without any obstruction, and switch reliably between its open and closed configurations. Three tests were defined to evaluate the robustness and performance of the valve: (a) closed configuration seal test; (b) switching test from closed to open configuration; and (c) switching test from open to closed configuration. In test (a), the valve (in manufactured seal mode) was examined for potential leaks at a spin rate of 6500 rpm for 4 min. No leakage of the sealed channel was observed as demonstrated in Figure 7a(i,ii). In test (b), the ferrowax blocking the channel in a normally closed wax valve was remelted by heating it with a laser (500 mW for up to 5 s). The disc was then spun at 6500 rpm for 4 min and after the wax moved into the downstream chamber, the fluid was transferred from the upstream chamber into the downstream chamber through the fluidic channel as seen in Figure 7b(i,ii). In test (c), the wax was reintroduced in the wax chamber inlet and heated. The molten wax driven by capillary forces blocks the fluidic channel and is allowed to solidify. After the wax is solidified in the channel, the inward chamber is filled with fluid and the CD is spun at 6500 rpm for 4 min (see Figure 7c(i,ii)). No leakage was observed, confirming a repeatable operation of the wax valve.

The number of times the wax valve can be used repeatedly to open and close the fluidic channel is limited only by the size of the wax reservoir and the ability of the downstream chamber to collect the removed wax.

For complex fluidic systems that have channels systems downstream such that when operating the wax valve some of the downstream channels could be obstructed, we introduce an indirectly connected wax valve that is placed further away from the valving channel. In Appendix A, we demonstrate one such design.

#### 3.1.2. Theoretical Analysis of the Solidification Time of Ferrowax Plugs

After the ferrowax is solidified in the channel (the closed configuration), it cuts off the fluidic pathway connecting the upstream and the downstream chambers. Theoretical analysis provides an estimate for the solidification time of ferrowax and the speed of switching from the on to the off position. In this way, we can determine the duration for which the wax remains molten, and thus for how long the fluidic channel remains open.

We estimate *t_d_* the wax solidification time, using Chvorinov’s rule [25]:(1)$$t_{d}=B{\left(\frac{V}{A}\right)}^{2},$$
where *V* is the volume of the wax in the channel, *A* is the area of the wax in contact with the surrounding acrylic, and the coefficient *B* is calculated using:(2)$$B={\frac{\rho_{m}L}{T_{m}-T_{o}}}^{2}\frac{\pi }{4k\rho c}\;\left[1+{\frac{c_{m}\Delta T_{s}}{L}}^{2}\right],$$
with the wax cooling rate ($c_{m}\Delta T_{s}$) approximately 1 W/g [26], the thermal conductivity (*k*) of the CD acrylic approximately 0.2 W/m⋅K [27], the melting point of wax ($T_{m}$) 65 °C , room temperature ($T_{o}$) of 25 °C, the latent heat of paraffin wax (*L*) 176 kJ/kg, the density of wax ($\rho_{m}$) 760 kg/m^3^, the density of the acrylic disc (ρ) 1180 kg/m^3^, and the specific heat of acrylic (*c*) 1.8 kJ/kg⋅C [28]. We calculate the parameter *B* to be approximately 6.9. With the volume of the wax *V* in the channel 4 × 10^−9^ m^3^, and the area *A* of acrylic in contact with wax 5 × 10^−4^ m^2^, Equation (1) yields a $t_{d}$ of approximately 10 s—which is lower than the experimentally observed wax solidification time of approximately 15 s. The difference between the predicted value and experimental observation is likely because the wax does not solidify uniformly, and the solidification of the wax is affected by the formation of air bubbles, which slow down the solidification of the molten wax due to reduced contact with the acrylic CD, which is at the room temperature.

Room temperature solidification of the wax plug takes approximately 15 s. In order to speed up the solidification of the wax, the spin station can have an on-board cooling unit such as a Peltier couple [20]. When we use refrigeration that emulates Peltier cooling, the wax solidification time is reduced to 5 s.

#### 3.1.3. Theoretical Analysis of the Dynamics of Opening of the Ferrowax Valve

To determine the lowest angular velocity to open a valve for emptying a dye/water from the upstream chamber into the downstream chamber, we equate the Young–Laplace equation’s pressure drop [29] across the molten wax meniscus,
(3)$$P_{c}=2\gamma cos\theta \left(\frac{1}{w}+\frac{1}{h}\right),$$
with the pressure exerted on the plug by the centrifugal forces,
(4)$$P_{r}=\frac{\rho \omega^{2}\Delta r\vec{r}}{A},$$

Here the pressure drop across the meniscus is $P_{c}$, the surface tension coefficient *γ* is 0.051 N/m, *θ* is the contact angle of wax/acrylic interface, which is approximately $17^{o}$, the width (*w*) and height of the channel (*h*) are both 0.5 mm, *ω* is the angular velocity of the disc, the average radial distance of the center of the wax plug in the valving channel is $\vec{r},$ the difference of the radial distances to the center of the disc for the top and the bottom meniscus of the molten wax plug in the valving channel is Δr (see Figure 7d), and the density of the molten wax plug is *ρ*. The capillary pressure in the valving channel was evaluated to be approximately 390.17 Pa. Using this value, we calculated the minimum angular velocity at which the removal of the molten wax is initiated to be approximately 121 rpm. Experimentally, the angular velocity required for the molten wax plug ejection from the channel is approximately 200 rpm. The higher experimental rpm can be explained by two factors: (a) the uncertainty of the physical parameters of the cooling wax; and (b) when the molten wax plug reaches the edge of the channel at the junction with the downstream reservoir, a capillary burst valve is formed and, subsequently, a higher centrifugal force is required to burst that valve.

#### 3.1.4. Theoretical Analysis of the Time to Melt the Wax

The energy required for phase transformation to melt the wax in the wax reservoir or *E_p_* if wax fills 50% of the wax reservoir (representing the approximate filling of the wax reservoir in our experiments) can be evaluated using:(5)$$E_{p}=L\rho V,$$
where *ρ* is the density of the ferrowax and the volume of the reservoir filled with ferrowax is *V*; the latent heat for ferrowax is *L* and is equal to 176 kJ/kg [30].

The value of *E_p_* required to melt the wax in the reservoir is calculated to be 0.535 J. The 500-mW laser used can deliver this energy in approximately 1 s. However, experimentally the time to melt the wax is closer to 5 s. This is because radiation is only partially absorbed [19]. In addition, because of the small spot size of the laser, the wax first melts only locally, and the rest melts through heat conduction. These trends are consistent with published results of phase-change microfluidic valves by Yang et al. [31].

As indicated in Section 3.1.2, it takes about 15 s for wax to solidify in the valving channel, so with passive ambient temperature cooling only (not employing active cooling with Peltier elements or other external cooling devices, such as fans), it takes approximately 20 s (5 s to melt the wax, followed by fast capillary wicking of the wax into the channel and 15 s for wax solidification) to close the valve. This time can be reduced if more powerful lasers and external cooling devices are used.

#### 3.1.5. Analysis of the Wax Plug Advancement in the Channel

We approximate the molten ferrowax advancing in the valve channel as capillary action inside a rectangular microchannel. In this case, the capillary pressure difference (*P_c_*) between atmosphere and the liquid meniscus inside the rectangular microchannel is given by the Young–Laplace’s Equation [29] (see Equation (3) above).

In microfluidic calculations, the hydrodynamic resistance *R_hyd_* [32] to fluid flow in a channel of rectangular cross-section of height h and width w is given by:(6)$$R_{hyd}=\frac{12\mu L}{h^{3}w\left(1-0.63\frac{h}{w}\right)},$$
where *μ* is the viscosity of the fluid and *L* is the wetted length of the channel. For capillary flow, the flow rate (Q) [32] is obtained by taking the time derivative of the volume whL (with only *L* changing with time as the wax flows):(7)$$Q\;=\;\mathrm{wh}\;\left(\frac{dL}{dt}\right)=-\frac{P_{c}}{R_{hyd}},$$

By solving this equation by assuming a shallow channel (*h* << *w*) and neglecting terms containing 1/*w*, we obtain the simplified expression for the flow velocity:(8)$$\frac{dL}{dt}=\frac{\gamma h\cos\theta }{6\mu L},$$

Upon integration of Equation (8), we obtain the so-called Washburn equation, relating the capillary advance with respect to time *t*. Here, the length of the capillary column *L* can be predicted to grow with time as:(9)$$L=\sqrt{\frac{\gamma ht\;cos\;\theta_{c}\;}{3\mu }}=W\sqrt{t},$$
where *W* denotes the Washburn constant that is a function of surface tension γ, the viscosity of the fluid *µ*, the contact angle of the fluid with channel walls $\;\theta_{c}$, and the depth of the channel *h*.

For the advancement of the ferrowax in the fluidic channel on the acrylic disc, the calculated value of *W* is 9.47 assuming the viscosity of the fluid *µ* = 0.00061 kg/m⋅s.

Figure 8a presents a plot that compares the experimental data for the timed advancement of the wax meniscus in the microfluidic channel (blue crosses) with the theoretical prediction of Washburn’s Equation (red line). These experiments are performed on a stationary disc, so centrifugal force plays no role, and the underlying physics is based solely on the capillary force. The plot initially coincides with the Washburn’s Equation predictions, but eventually the capillary flow experimentally observed is slower than the theoretical prediction. This deviation likely relates to the cooling of the molten ferrowax and thus to the change in its physical properties. This can be explained by the increase of viscosity μ (due to the cooling effect of the acrylic at room temperature). The simulation predictions of the molten wax location (Washburn’s Equation) are within 18.75% of the error margin with respect to experimental results.

#### 3.1.6. Effect of Valve Convergence Angle on Response Time of Wax Capillary Valve

An important parameter that affects the performance of the capillary-based ferrowax valve is the valve convergence angle (see Figure 3b for its definition). Keeping the other valve parameters constant, the valve convergence angle is studied to understand its effect on the dynamics of the movement of the molten ferrowax from the wax reservoir to the valve channel. In this study, wax reservoirs with five different convergence angles: 20°, 25°, 30°, 35°, and 40° were compared. In each test, the same mass of solid ferrowax was placed in the wax reservoir in such a way that the wax edge was 3 mm away from the entrance to the fluidic channel. The solid wax was then melted using a laser (500 mW for up to 5 s) to initiate the capillary flow. Figure 8b presents a plot of the time it takes for the molten ferrowax to flow into the fluidic channel. It is observed that the time required for the wax to reach the channel is decreasing with an increase in the convergence angle of the sidewalls of the wax reservoir. Similar behavior was observed in capillary burst valves (CBV) by Chen et al. [29], where the wedge angles show a similar relationship with the burst frequency of the CBVs. This is explained by Equation (6) where the geometry of cross-section is inversely proportional to the flow resistance. With higher convergence angle, the average cross-section area of the wax reservoir increases. Thus, this variation lowers the resistance and therefore lowers the response time.

#### 3.1.7. Effect of Valve Area of Capillary Path

Another prominent factor that affects the Valve closure time is the cross-section of the channel to be closed. As with the valve convergence angle, we measured and plotted the closure times for different cross-section areas of the valving channel. We used ANSYS Fluent to measure the valve closure time and used the Solidworks model with different areas of the capillary path as an input into the simulation. We plotted the simulated values in Figure 8c as a solid red line, which is a straight line like our solid wax curve. We used the heated Hamilton syringe setup to measure the valve closure time and used valves with different areas of the capillary path. We let the molten wax enter the wax opening using the Hamilton syringe and measured the valve closure time. The measured values are plotted in Figure 8c as blue crosses and are approximately linear. We concluded that as the valve’s area of capillary path increased, it required more time for the capillary flow to close the channel. Similar results are also documented in this paper by Yang et al. [31]. We can use this relationship to properly design the wax valve for a particular application and time response.

### 3.2. Operation of the Magnet-Actuated Ferrowax Valve

#### Functional Testing of the Magnet-Actuated Ferrowax Valve

Similar to the capillary wax valve (see Section 3.1.1), the operation of a magnet-actuated ferrowax valve was evaluated using three validation tests: (A) closed configuration seal test; (B) switching test from closed to open configuration; and (C) switching test from open to closed configuration. In test (A), the valve is closed by heating the wax in the wax reservoir, using the magnet to move the wax into the valving channel, and cooling to solidify the seal. To examine for potential leaks of the valve, the disc is spun at 6500 rpm for 4 min and no leakages were observed, demonstrating the sealing of the channel. In test (B), the wax-sealed fluidic channel is opened by heating the ferrowax and then either spinning the CD at 6500 rpm to propel the molten wax from the channel into the downstream reservoir or using a magnet to reverse the flow of the ferrowax back into the wax reservoir. Finally, in test (C), switching from open to closed configuration is tested by heating the ferrowax in the wax chamber using a laser (500 mW for up to 5 s). The magnet is advanced from the top of the wax chamber towards the valve channel, which pulls the molten ferrowax back and blocks the fluidic channel. Subsequently, the CD is cooled (by placing it in the refrigerator for 5 s at 4 °C), solidifying the wax and closing the channel. This closed configuration is tested for leakages, by spinning the disc at 6500 rpm for 4 min. No leakages were observed, demonstrating the passing of the validation test (C) (see Figure 1c,d for images of the magnet wax valve operation).

Finally, we find the response times (without melting and solidification) of the capillary wax valve is approximately 5 s and the magnetic wax valve is approximately 1 s.

## 4. Conclusions

We have demonstrated two new types of wax valves for application in centrifugal microfluidics. Solid ferrowax is held in a wax reservoir and is melted either by a hot plate beneath the stationary disc or by a laser. Once the wax is molten, it flows into the valve channel and upon solidification it cuts off the flow between the inner and outer reservoirs that are connected by the valve channel. Aside from purely capillary flow, it is also possible to transfer the molten ferrowax from the wax reservoir into the valve channel by an external magnet that forces the ferrowax to flow towards the valve channel. It was demonstrated experimentally that, whereas it takes approximately 5 s for the laser to melt the wax, it takes approximately 15 s for the wax to solidify to close off the fluidic channel. We have performed a series of simulations of the capillary wax valve where the valve switches from the normally open to closed configuration and the normally closed to open configuration, which show good agreement with the experimental observations. Additionally, a theoretical analysis of the wax melting, solidification, and movement of the wax in the fluidic channel under the action of capillary force was performed. The discrepancy between the theoretical analysis and experimental results is possibly due to the changing physical properties of the cooling wax (changing viscosity), the surface roughness of the acrylic CD, and machining tolerances. We have also performed experimental studies of the role of the converging angle of the wax reservoir on the kinetics of the molten wax flow towards the fluidic channel and discovered that a larger converging angle causes faster flow of wax towards the valve channel. For the magnetic wax valve, we demonstrated reversible opening and closing of the valve by performing validation tests to check their sealing and switching capability.

Experimental results demonstrated the reversibility, multi-use, and reliability (seal-proof status) of the ferrowax valves. The active wax valves described in this work, although not the fastest active microfluidic valves, have a distinct advantage that they can be used to hermetically seal and reseal chambers to afford long-term storage of reagents. These novel ferrowax valves can be used to open and close fluidic channels on demand and to seal off chemical and biological reagents or samples on the centrifugal microfluidic platforms and thus be implemented for future point-of-care diagnostics. A suitable example of application of the presented multiuse wax valves is that of integration of blood-plasma separation with biosensing array application. In such a CD (already developed in BioMEMS lab), we first perform blood plasma separation and collect the red blood cells in an isolation chamber which is closed using a wax valve. The remaining plasma is used for biosensing array using reciprocation on CD. The red blood cells can be removed from the isolation in the chamber (reopening the valve) and used for further analysis on the CD.

## 5. Patents

A patent has been filed at UC Irvine under patent number UC Case No: 2021-781-1.

## Figures and Tables

**Figure 1 micromachines-13-00303-f001:**
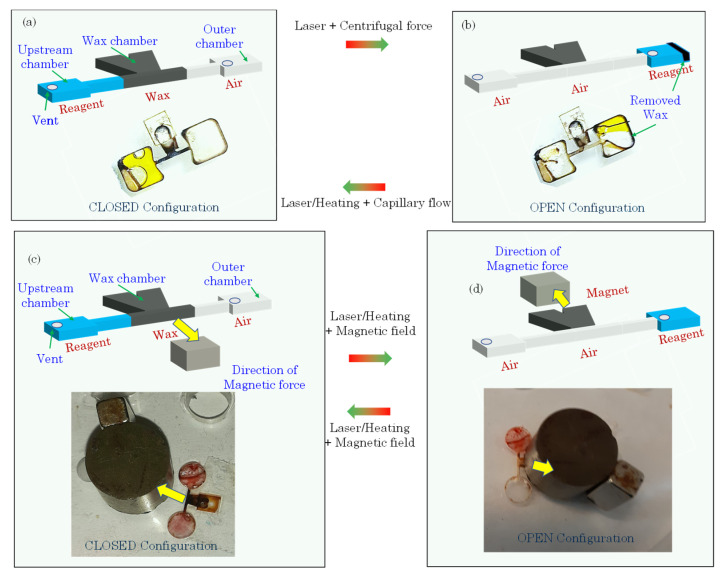
(**a**) Normally closed configuration of the capillary wax valve. It can be switched to open mode by melting the wax plug blocking the channel (using a laser) and spinning the disc to flow the molten wax plug downstream, clearing the channel to open the fluidic channel. (**b**) The switch from normally open to closed configuration is achieved by melting the ferrowax in the wax chamber using a laser. Under the action of the capillary force, the molten wax flows into the valving channel and seals it, thus switching the valve from normally open to closed configuration. With enough wax in the wax chamber, this process is repeatable. (**c**) Normally closed configuration of the magnetic wax valve. It can be switched to open by melting the wax plug blocking the channel and the wax reservoir (laser/heating) and moving the wax with the magnet out of the channel, clearing the channel and opening the valve from normally closed to open configuration. (**d**) The switch from normally open to closed configuration (magnet wax valve) is achieved by melting the ferrowax in the wax chamber (using laser) and moving the wax with the magnet into the channel. This reversibly changes the configuration from normally open to closed configuration. Yellow arrows show the direction of the magnetic force.

**Figure 2 micromachines-13-00303-f002:**
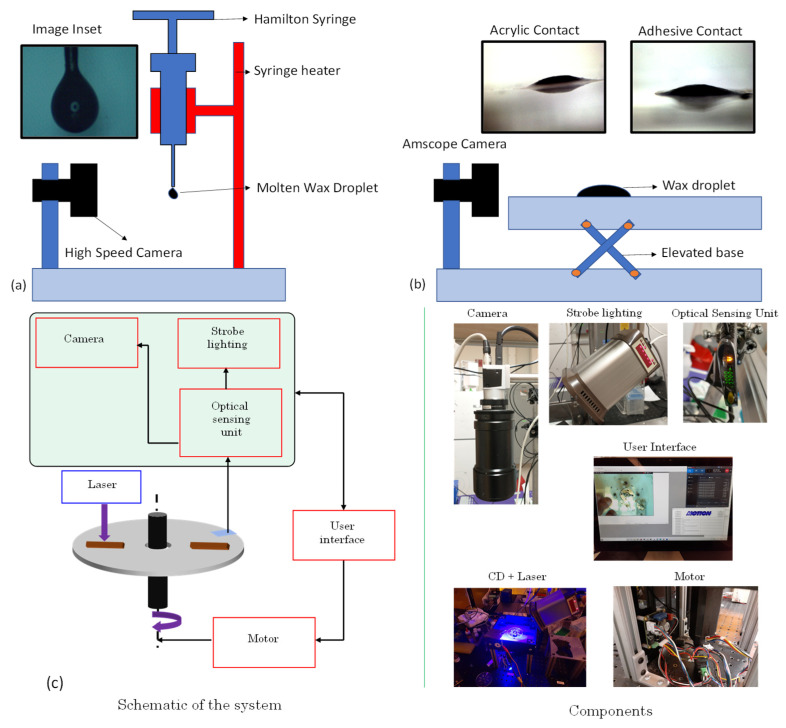
Ferrowax property measurement and characterization setup: (**a**) Viscosity measurement setup: High-speed camera Kron Technologies 18W15 (Burnaby, BC, Canada) takes images of pendant drops of the molten ferrowax (see inset) suspended from the needle of the heated Hamilton syringe. Images of the droplets are processed via ImageJ Pendant Drop Plugin [21] to determine the surface tension of the ferrowax. (**b**) Contact angle measurement setup: Amscope MA1000 camera (Irvine, CA, USA) takes pictures (see insets) of the ferrowax drops on various surfaces, such as on adhesive film and acrylic plastic (inset) and images are processed with ImageJ software (contact angle plugin) [22] to determine the corresponding contact angles of ferrowax with different surfaces. The measured values are listed in Table 1. (**c**) Experimental setup for testing and imaging of the wax valve experiments. The motor (Smartmotor SM3450D, Motion USA, Columbus, OH, USA), rotating the spin chuck (lab machined), is connected to a controller (EZSV23/EZSV17, AllMotion, Union City, CA, USA) in which users enter the angular velocities and accelerations of the discs via a computer user interface. The actual angular velocity is measured using the light backscattered from a reflecting tape strip placed onto the disc (camera’s shutter as well as the strobe light are also actuated using this reflected light). The camera images are recorded using a screen grabbing software called Bandicam for analysis.

**Figure 3 micromachines-13-00303-f003:**
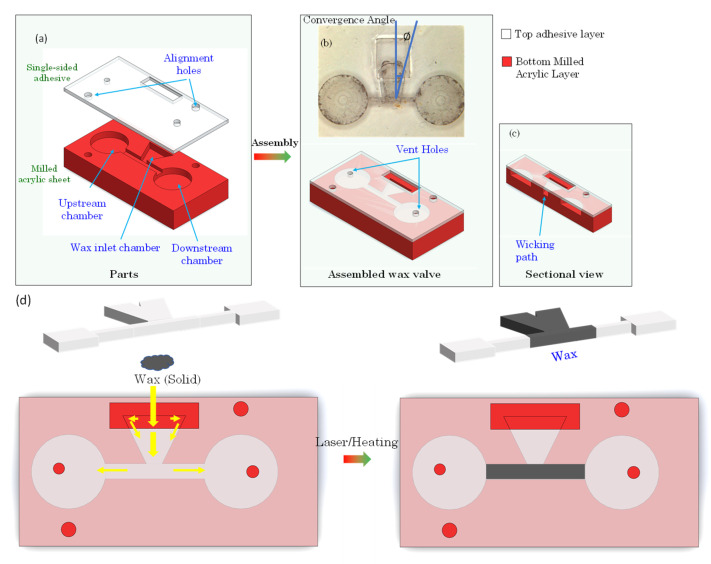
(**a**) Layers of the wax valve: the top layer of this assembly consists of a single-sided adhesive sheet that contains the openings for vent holes, alignment holes, and openings for loading the wax in the wax chamber; the bottom layer is an acrylic layer machined, according to the valve 3-D model, to provide for the upstream chamber, the wax inlet chamber, the fluidic channel, and the downstream chamber. (**b**) The layers are pressed together using a roller (Akiro, B07YDNKSH6) to activate the pressure-sensitive adhesive that binds the layers together. Top diagram also defines the valve convergence angle, which is the angle between the perpendicular to the valving channel and the inclined side of the wax chamber. (**c**) Cross-sectional view of the valve showing the wicking path of the wax. (**d**) Wax is deposited as a solid pellet through the opening in the adhesive layer (left). After heating, the molten wax under the influence of capillary force (yellow arrows) flows into the fluidic channel and blocks the fluidic path (right). Red circles and rectangles indicate the openings in the top layer. Larger red circles denote the alignment holes.

**Figure 4 micromachines-13-00303-f004:**
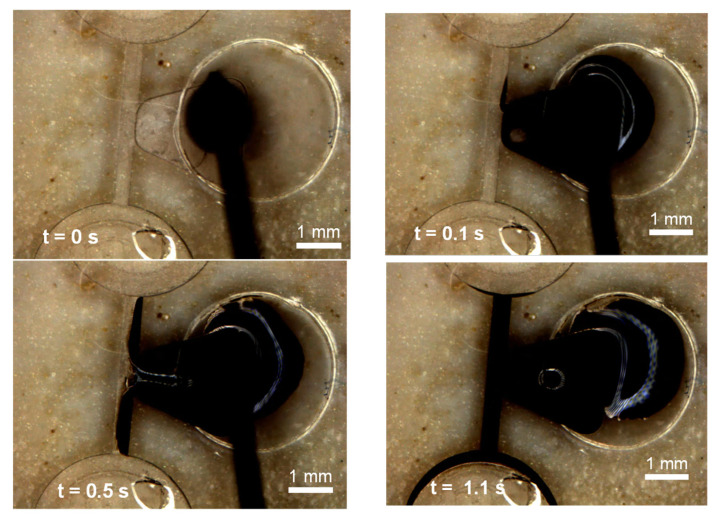
Images (with the corresponding time stamps) showing the deposition (using a heated Hamilton syringe) of the molten ferrowax and its subsequent motion due to the action of the capillary forces and the blocking of the fluidic channel.

**Figure 5 micromachines-13-00303-f005:**
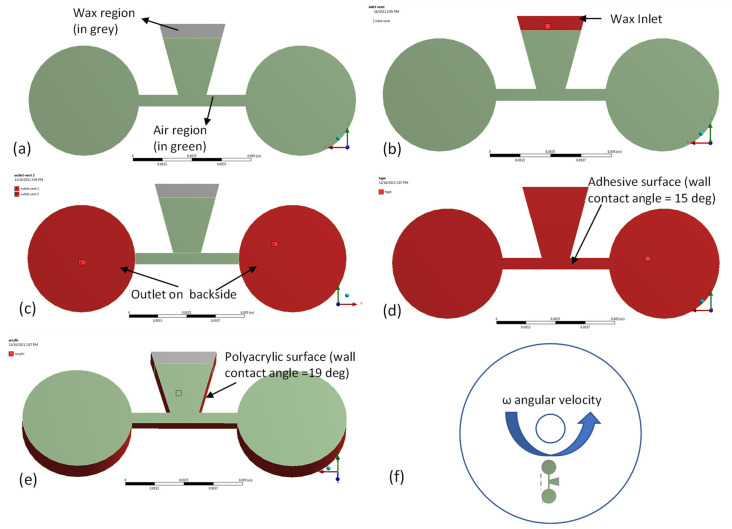
(**a**) Simplified 3D geometry used in the ANSYS simulation with the molten wax region represented by grey (volume fraction (VF) ferrowax = 1, VF air = 0) and the air region represented by green (VF ferrowax = 0, VF air = 1). We use different colors to show two different phases and their presence region-wise. (**b**) The location of the inlet for the molten wax (VF ferrowax = 1, in red); (**c**) the location of the outlet (the 3D model is flipped to the backside) on the bottom of the two fluidic chambers (in red); (**d**) the adhesive layer (in red) forms the top surface on the wax chamber and the valve channel; (**e**) the location of the acrylic surface (in red); (**f**) the position and orientation of the wax valve on a disc and the direction of rotation of the CD.

**Figure 6 micromachines-13-00303-f006:**
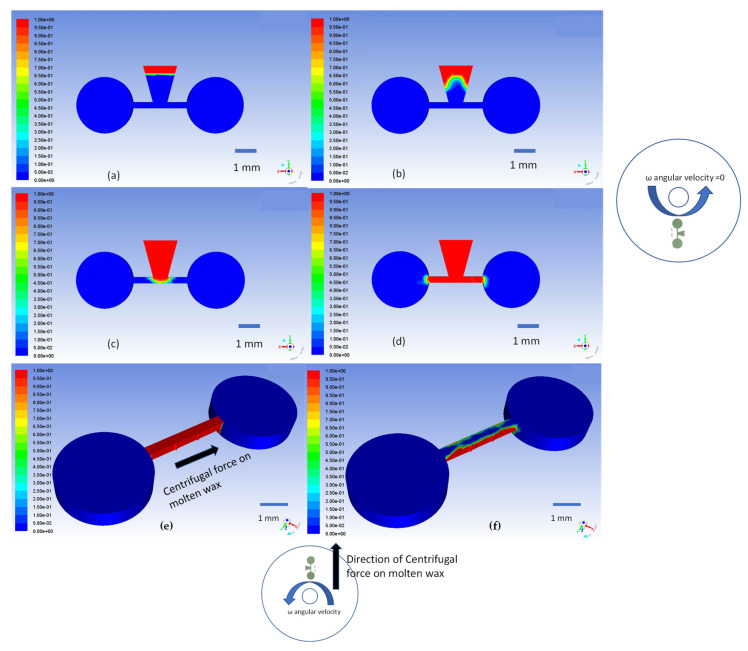
(**a**) The simplified 3D geometry used in ANSYS simulation of two-phase laminar flow represents the molten wax region in red (V.F wax = 1) and the air phase in blue (V.F wax = 0); (**b**) the air/wax interface is the concave meniscus moving towards the valving channel; (**c**) moving molten wax is entering the valving channel; (**d**) the molten wax fills the channel and blocks the connecting channel between the inward and the outward chambers. The right illustration depicts the location of the valve and the spin direction of the CD. (**e**) ANSYS simulation of the normally closed to open valve with the molten ferrowax blocking the connecting channel. A centrifugal force for spinning of the CD at 6500 rpm is applied. This is also shown in the inset of the CD at the bottom. (**f**) The simulation results clearly show that the centrifugal force pushes the molten wax towards the outward channel and opens the connecting fluid channel. The color indicates the volume fraction of molten wax (color: red, VF wax = 1.0) vs. air (color: blue, VF wax = 0.0).

**Figure 7 micromachines-13-00303-f007:**
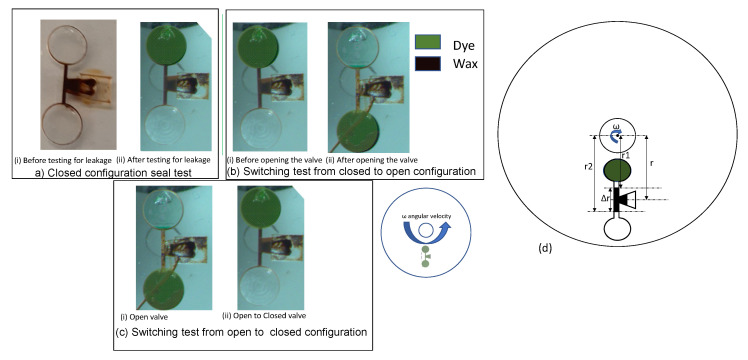
Experimental results after the functional testing of the capillary force-based wax valve: (**a**) closed configuration seal test: test is performed on a normally closed configuration wax valve shown in (**i**) by spinning the CD at 6500 rpm in (**ii**) and checking for leakage in the lower chamber; (**b**) switching test from closed to open configuration: The wax plug in the valve channel as shown in (**i**) is melted and the disc is spun at 6500 rpm upon which the molten wax flows to the outward chamber under the influence of the centrifugal force thus enabling the fluid (green dye) to move to the outward channel as shown in (**ii**); (**c**) switching test from open to closed configuration: The valve in the normally open configuration (**i**) is heated in the wax reservoir to melt the wax and the molten wax (again under the influence of the capillary force) flows to the channel to block the valving channel and is cooled for 15 s to solidify and seal the channel. No leakage was found (**ii**) after spinning the disc at 6500 rpm for 4 min. (**d**) Schematic of the location of the elements of the capillary wax valve on a centrifugal disc platform with respect to center of the disc. Green shaded region denotes the fluid location and black shaded region denotes the ferrowax.

**Figure 8 micromachines-13-00303-f008:**
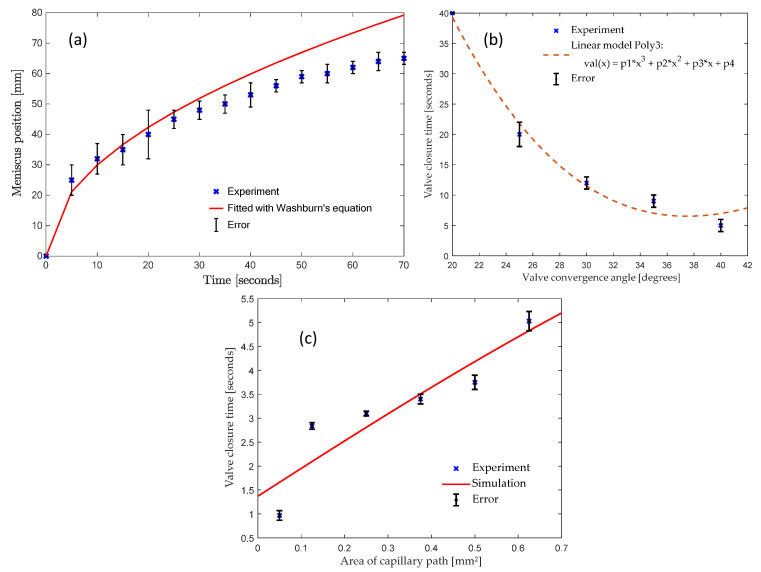
(**a**) Plot of the time of vs. advancement of the molten ferrowax meniscus in the fluidic channel under capillary action. The experimental values are indicated by blue cross marks (where the error bars represent one standard deviation). The red line indicates the trend corresponding to Washburn’s Equation (Equation (9), with W = 9.47). The simulation predictions of the molten wax location (Washburn’s Equation) are within 18.75% of the error margin with respect to experimental results. (**b**) Plot depicting the relationship between the valve convergence angle (see Figure 3b for definition) and the time it takes for the molten wax to reach the fluidic channel. The error bars represent one standard deviation. Dotted line is a fitted curve to the blue crosses. (**c**) Relationship between the valving channel cross-section and the response time. The red line shows the simulated curve and blue crosses represent experimental measurements done using the Hamilton syringe setup where we use a Hamilton syringe to input the wax.

**Table 1 micromachines-13-00303-t001:** Properties of ferrowax used in the ANSYS simulation.

Property Input Type	Value with Units
Density of ferro-wax [23], ρ	760 kg/m^3^
Surface tension of ferro-wax at 90 °C *, γ	0.051 +/− 0.005 N/m
Viscosity of ferro-wax at 90 °C [23], μ	0.00031 kg/m⋅s
Contact angle (acrylic surface) * ferro-wax, θ_c_	19 +/− 0.5°
Contact angle (adhesive) * ferro-wax, θ_c_	15 +/− 0.5°

* Measured as discussed in Section 2.

## Data Availability

Not applicable.

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
