# Peer review of "Capillary Flow-Driven and Magnetically Actuated Multi-Use Wax Valves for Controlled Sealing and Releasing of Fluids on Centrifugal Microfluidic Platforms"

_micromachines, 2022, doi:10.3390/mi13020303_

Round 1
Reviewer 1 Report
The authors demonstrated two new types of wax valves for application in centrifugal microfluidics. The solid ferrowax is stored in a wax reservoir, and the microchannels are opened or closed by heat. Another way is to apply a magnetic wax valve. Both new types of wax valves presented by the author use the “active” method. As the author also noted in the “Introduction” section, active methods are generally characterized as being faster, more accurate, or with higher precision than passive methods. However, it is difficult to say that the two methods presented by the author have fast operation or high precision in microchip utilization. Is there an important reason to use a special heat or magnet? Also, why is it important to rotate the Centrifugal microfluidic platform in the experimental section? (Is it correlated with capillary force?)
Reviewer 2 Report
The manuscript describes two approaches to achieve a wax valve for implementation on lab-on-cd systems. The basic idea is that molten wax mixed with ferrofluid can be made to flow into a channel via capillary forces or magnetic forces and then allowed to set. This blocks the channel and constitutes the closed valve configuration. Heating the wax and using centrifugal forces or magnetic forces to remove it from the channel put the valve into the open configuration. Several approaches are described and the valve can be set into the open or closed configuration using the approaches.
The method seems to show promise and would indeed be very useful when fully optimized. However there appear to be some discrepancies between what is promised in the title and abstract and what is actually shown in the manuscript. My biggest reservation is for the multi-use aspect. I do not feel that the authors have fully shows the valve being used for multiple cycles. This can be remedied by either changing what is promised in the title and abstract, by a clearer description of what is meant by multi-use or by showing more clearly how the valve has been multiple times. For example, what is meant by multiple?
Another issue I have with the manuscript is the fact that much of the switching is done in the absence of the reagent. The surface tension of the wax is considered in air, the wetting angles of the wax are considered in air, the capillary wetting of the channel is considered in air. The simulations of the capillary filling of the valve channel with wax are also done in air. For a one-shot valve this seems perfectly ok, but one of the major claims in the manuscript is that the valve is reusable. Surely a fully reusable valve should be functional in the presence of the reagent in both the on/off transition and the off/on transition. The authors have not shown any proof that this is possible.
Also relating the reusability: the authors state on line 312 that the number of times the valve can be reused depends only on the size of the wax reservoir. The wax must go somewhere though. A reference is made to this and to another design in the following sentences but this should be made clearer. The unfortunate choice of colours in figure 7 means that it is not possible to distinguish between the wax and the reagent.
The need to place the system in a refrigerator to solidify the wax seems to be a limitation if the system is to be easily reusable as a point-of-care device. It would be interesting to here in which way the authors envision the reusability being taken advantage of. Would they be refilling a device in order to save resources? Or using the switching to perform multiple steps in an analysis? The slow switching times would seem to make the first possible but they would have to be improved considerably to make the second approach doable in all but very slow assays. Some examples might be e helpful for the reader.
While the simulations confirm the fact that the capillary force causes the wax to flow into the channel under the conditions that are present, this is not really at question. A much more interesting question to answer with the simulations would have been: how could the design be improved. The authors tested actual devices with different angles but they could have tested this using the simulations. The authors could also have varied the dimensions of the valve channel in the simulations to optimize the melting, setting, filling and emptied times since these are all important to the switching of the valve.
In the derivation of the Washburn equation for the channel the authors use the common approximation for the hydrodynamic resistance in a rectangular channel. However, this only holds for w>>h and the channel in the device presented in the manuscript has w=h (line 284). This might help to explain why the fit in figure 8a is poor. It is also unclear which value for the viscosity is used. This is temperature dependent but the authors do not discuss which temperature is reached in the device. In table 1 the authors give the value for 90 degrees C. Is this the value used in the Washburn equation and if so how is it motivated?
In many places in the manuscript the authors state that the wax was melted by placing the device on a hotplate or by the use of a laser. However, it not always clear which was used. It would be easier to read if the author stated which was used to get the results that are presented and refrain otherwise from generalising too much.
The numbering of the figures (eg. 7Aa) is unnecessarily confusing. 7a i and ii would be better.
Figure 3d, the figures are not fully visible.
In the text to Figure 8b a fitted curve is mentioned but not the form of the function that is fitted.
Reviewer 3 Report
In this work, Peshin et al. have demonstrated new designs for wax-based valves to control flow (sealing and releasing) in centrifugal microfluidics. The reading of the manuscript was comfortable, the message was clear, and the content of the introduction was appropriate. Although ferrowax valves had been used in centrifugal platforms, this one presents two alternative simple approaches: 1) wax-valve based on heating (hotplate or laser) to melt the wax that flows to the valve channel and closes it after cool down, and 2) reversible magnetic wax valve using an external magnet to control the position of wax barrier. The first approach using capillary force is incremented by simulation by finite element analysis. Some theoretical analyses were also performed and compared to experimental conditions. I do recommend the publication of this manuscript as it is. The comments below are just to improve the quality of the manuscript.
- Please, show and indicate vent holes in Figure 1.
- In Fig 1, after heating and rotation, it looks like melted wax has disappeared. Indicate the molten wax final position.
- In Fig 1, increase the size of the photography showing the magnet position.
- Figure 4 and Video 2 (Supp Info): when molten wax was introduced to the chamber it is possible to see the wax instantaneously leaking to both upstream and downstream chambers. Is it possible to avoid it by optimizing experimental conditions as the volume of the injected molten ferrowax, or by adjusting the channel size (forming a capillary burst valve)? Please, include a statement discussing how this can be experimentally optimized or avoided for future applications.
- Please, compare these new types of valves in terms of advantages/applications to previous published ferrowax-based reversible valves (Park et al. LabChip, 7, 557, 2007).
Minor point:
Page 13, line 401: verify ref 29 presented as superscript instead of [x] form.
Round 2
Reviewer 2 Report
Dear authors, thank you for considering my comments so thoroughly. I believe that the revised manuscript is considerably better based on the changes made in response to the concerns of both myself and the other two reviewers. I would be very happy to see the manuscript published.